# Effect of Aging on Unidirectional Composite Laminate Polyethylene for Body Armor

**DOI:** 10.3390/polym15061347

**Published:** 2023-03-08

**Authors:** Amy Engelbrecht-Wiggans, Zois Tsinas, Ajay Krishnamurthy, Amanda L. Forster

**Affiliations:** 1Mechanical Engineering Department, Rochester Institute of Technology, Rochester, NY 14623, USA; 2Material Measurement Laboratory, National Institute of Standards and Technology, Gaithersburg, MD 20899, USAamanda.forster@nist.gov (A.L.F.); 3Theiss Research, La Jolla, CA 92037, USA

**Keywords:** composite laminate, flexible composite, strip tensile testing, body armor, polyethylene, hydrothermal aging

## Abstract

The construction of ballistic-resistant body armor is experiencing an increasing use of flexible unidirectional (UD) composite laminates that comprise multiple layers. Each UD layer contains hexagonally packed high-performance fibers with a very low modulus matrix (sometimes referred to as binder resins). Laminates are then made from orthogonal stacks of these layers, and these laminate-based armor packages offer significant performance advantages over standard woven materials. When designing any armor system, the long-term reliability of the armor materials is critical, particularly with regard to stability with exposure to temperature and humidity, as these are known causes of degradation in commonly used body armor materials. To better inform future armor designers, this work investigates the tensile behavior of an ultra-high molar mass polyethylene (UHMMPE) flexible UD laminate that was aged for at least 350 d at two accelerated conditions: 70 °C at 76% relative humidity (RH) and 70 °C in a desiccator. Tensile tests were performed at two different loading rates. The mechanical properties of the material after ageing demonstrated less than 10% degradation in tensile strength, indicating high reliability for armor made from this material.

## 1. Introduction

Modern technological advancements in weaponry and ammunition, as well as wearer requirements for comfort and mobility, have led to the development of new body armor materials that are flexible and light weight, with increased damage resistance and better energy absorption capacity [1]. Many light-weight ballistic protective armor systems are currently constructed using high modulus and high strength polymeric fibers such as ultra-high-molar-mass polyethylene (UHMMPE) [2,3] and poly(p-phenylene terephthalamide), or PPTA [4,5], commonly known as aramid, and new composites are also quickly emerging [6]. In order to ensure the wearer’s safety over the armor’s expected use lifetime, the development of these new materials and their use in body armor applications requires a deep understanding of their long-term stability and potential degradation mechanisms. Historically, little emphasis was placed on the long-term performance of armor materials. However, there was a renewed effort to understand the body armor failure mechanisms in the wake of an armor failure in 2003 [7]. Significant efforts have been launched to examine the mechanisms of aging in UHMMPE and PPTA fibers.

There are several advantages to flexible UD composite laminates compared to traditional woven material. Flexible UD laminates have been shown to absorb more energy per unit panel weight [8,9,10], with similar applications and flexibility as woven fabrics [11]. In addition, UD composite laminates with UHMMPE or PPTA fibers have been shown to perform significantly better in out-of-plane compression compared to woven materials [12]. Either UHMMPE or PPTA fibers can be used in combination with an elastic resin material (20% or less by mass) [13] that can penetrate to the filament level and hold the fibers together to form a non-woven unidirectional (UD) tape. Layers of the unidirectional tape are laminated together in a crisscross pattern, where each layer is placed at a 90° angle from the layer above and below to form a panel, as depicted in Figure 1, where there are four such UD layers. Unidirectional fiber composite materials can vary significantly based on their intended application. Composites developed for aerospace applications can be hundreds of fibers thick with a high matrix volume fraction up to 50% [14], whereas in body armor applications, each layer has a thickness of less than 10 fibers (as can be seen in Figure 1) with lower matrix volume fractions due to the near-hexagonal packing arrangement of the fibers [9].

The degradation of UHMMPE fibers and the subsequent decrease in their mechanical properties under normal use conditions, involving elevated temperature, humidity, or some combination thereof, can occur through a well-documented process known as thermal oxidation [2,15,16,17,18,19,20,21,22,23]. Elevated temperatures can induce C–C bond scissions along the backbone of a UHMMPE chain, primarily in the amorphous regions of the polymer [24]. The rupture of these bonds results in the formation of C-centered free radicals (alkyl radicals) that can undergo various reactions which, in general, depend on the presence of oxygen in the amorphous areas, the concentration of oxygen present, the temperature, the moisture content, the presence of antioxidants in the material, and the degree of crystallinity [3,24,25,26]. In the presence of oxygen solubilized in the amorphous regions of UHMMPE, alkyl free radicals will be involved in a series of reactions that can further decrease the molar mass of the polymer and result in products such as ketones, alcohols, and water, as shown in Figure 2 [3,24,27,28]. These products will give the polymer hydrophilic properties, to a certain extent, making it more susceptible to moisture, which can, in turn, penetrate the polymer and induce more oxidation over time.

Compared to the UHMMPE yarns, only preliminary research has been performed to characterize the effect of environmental exposure on the matrix used in the UD laminates. Furthermore, the effect of changes in the binders on the ballistic performance of the UD laminate is unclear as the impact force is carried mainly by the fibers. A computational study investigating the ballistic performance of UHMMPE UD laminate composites, however, has shown that filament/matrix de-bonding/cohesion is of paramount importance in the performance of these materials and one of the main failure parameters upon impact [29]. Hence, it is prudent to test the entire composite structure to comprehensively determine the effects of aging. The current study evaluates the tensile strength of a UHMMPE flexible UD laminate under two sets of environmental aging conditions. Of key interest is how the *V*_50_ velocity, the velocity at which half of the projectiles penetrate, changes with aging. The *V*_50_ can be related to the tensile strength through empirical research by Cunniff [30] and theoretical research by Phoenix and Porwal [31], as detailed in [5,32]. Furthermore, tensile strength was chosen as a relevant metric because it is less complicated to measure than *V*_50_ and requires less material.

## 2. Materials and Methods

### 2.1. Material and Specimen Allocation

In this study, the flexible UD composite laminate comprised four layers, each layer having an approximate thickness of three UHMMPE fibers, where each fiber has an approximate diameter of 11 µm, all held together with a very low modulus matrix. When received, the laminate had damage spread sporadically throughout the material. Given the limited availability of the material, it was not possible to avoid the damaged material for the purposes of this study. Therefore, specimens were first cut from all of the available material, then each specimen’s level of damage was visually assessed and rated on a scale of 1 to 5. In particular, the types of damage seen included folds, spots of foreign material, areas where the fibers were not as clearly defined, and dirt that had become embedded. These factors were assessed as a whole to group specimens into the following categories: ‘not damaged’, ‘minor damage’, ‘minimal damage’, ‘moderate damage’, and ‘major damage’. Specimens were then randomly sorted into batches, controlling for the damage levels, so that each batch contained approximately the same number of specimens for each damage level. The specimens were about 70 mm wide with a 250 mm nominal gauge length, which is the same gauge length used in a previous UHMMPE yarn aging study [2]. Specimens were cut using a medical scalpel and a straight edge on a self-healing cutting mat, taking care to align the specimens with the fiber direction. The specimen preparation procedure is fully described in [33]. Specimens were cut with their length nominally perpendicular to the axis of the material roll, i.e., such that they would be along the warp fibers if the material was woven. 

The thickness was determined using measurements of the bulk unaged material at 18 locations throughout the roll using an electronic micrometer with a friction thimble (resolution of 0.001 mm and precision of 0.002 mm). The average thickness was 0.199 mm with a standard deviation of 0.011 mm from 10 measurements. Figure 3 shows a typical scanning electron microscopy (SEM) micrograph of the cross-section. For the purposes of this work, all tensile strength values are given as the failure load, rather than as stresses. This is because we are not able to calculate the fiber volume fraction accurately for this material, and nominal composite stresses (using the whole cross-sectional area, including cross fibers that carry no load) are not highly relevant. When the fiber volume fraction is known, then the fiber stress can be calculated from the nominal composite stress through the rule of mixtures. For a composite with such a large difference in modulus between the binder matrix and the fibers, the fiber stress can be approximated by dividing the load by the cross-sectional area of fibers parallel to the direction of the applied load. Approximating the UHMMPE fiber cross-sectional area is challenging in these laminates, particularly because the fibers are not cylindrical. This is further complicated by the amount of distortion cutting causes, as seen in Figure 3, where it is not possible to see how much space is between the fibers. The matrix fraction for these laminates is typically less than 20% by mass [5,13].

### 2.2. Accelerated Aging Parameters

After sorting the specimens into randomly assigned batches, they were aged in an environmental chamber at 70 °C with 76% RH and 70 °C in a desiccator. This temperature was chosen to accelerate the aging process as much as possible without inducing new forms of degradation, such as combustion, based on prior studies of UHMMPE [2]. The 70 °C temperature was also of interest for ready comparison to prior studies of UHMMPE yarns at 65 °C in a desiccator [2]. Some studies on UHMMPE have shown that humidity may be a further source of degradation [3], so, for this study, it was decided to age the material at 70 °C in both a humid and a dry environment. For the humid environment, 76% RH was chosen as it has been used for other aging studies on ballistic materials. The environmental chamber provided control to ±1 °C and ±5% RH.

Five extractions were performed at approximately 14, 70, 150, 230, and 350 d. After extraction, the specimens were stored overnight (or longer) so that they had time to equilibrate to laboratory conditions before testing. All testing was conducted at room temperature with mean laboratory conditions of 22 °C and 51% RH (standard deviations are 0.5 °C and 1.4% RH from 17,701 observations).

### 2.3. Mechanical Testing

The specimens were tensile tested in a screw-driven universal load frame equipped with a 30 kN load cell using capstan grips. Testing was performed at two different crosshead displacement rates (2 and 250 mm/min), with the faster loading rate chosen to match the previous yarn aging study and the slower rate to match a similar study on PPTA laminates. Strain was measured using a non-contacting video extensometer, with measurements taken at three different locations along each specimen’s width, as can be seen in Figure 4. These three strain values were averaged to determine the failure strain. Further details and rationales on the specimen aging, cutting, and testing procedure are described in [33]. Figure 4 shows a specimen before and after mechanical testing. Videos of all the tested specimens are available in the associated data publication [34].

One unaged specimen and one specimen from the longest aged humid and dry extraction were cyclically loaded in accordance with the following regime. At a crosshead displacement rate of 20 mm/min, the load was increased to 500 N and then decreased to 50 N ten times, with no pause at either 500 or 50 N. Then, the load was increased to 1 kN more than the previous maximum and decreased back to 50 N for another 10 cycles, i.e., increased to 1.5 kN, decreased to 50 N, increased to 2.5 kN, decreased to 50 N, etc. After the 20th decrease to 50 N, the specimen was loaded to failure. The same grips and video extensometer were used for both the cyclic loading and the tensile test. The loading rate of 20 mm/min was selected for the cyclic testing as an order of magnitude between the slow and fast (2 mm/min and 250 mm/min) loading rtes used in the tensile testing. The tensile loading rate of 250 mm/min rate was not used for the cyclic tests as it was too fast for careful data collection, particularly at the points where the crosshead changed direction. In contrast, with the 2 mm/min loading rate, the cyclic loading procedure takes long enough that creep of the material becomes an issue. Thus, 20 mm/min was chosen as a compromise between the two tensile test rates.

### 2.4. Scanning Electron Microscopy

Scanning Electron Microscopy (SEM) imaging was performed on the unaged fiber laminate after depositing a nominal 3 to 5 nm Au/Pd coating on a small piece of laminate (less than 0.5 mm by 0.5 mm by 1 mm). The imaging parameters were kept identical to the previous studies described in [5,35]: magnifications ranging between 50× and 3500× are typically used for imaging at a nominal working distance of less than 10 mm. To prevent electron-beam-induced fiber damage and reduce any charging artefacts, all imaging was conducted at accelerating voltages from 2 to 5 kV and currents from 50 to 200 pA.

## 3. Results

All data associated with this publication is archived through the NIST Public Data Repository [34]. Specimens of the UD material were cut 70 mm wide, aged at two different humidity levels for the periods previously described, and tested at two different crosshead displacement rates. Figure 5 is a plot of the average failure load, also referred to as the strength, with at least 50 specimens per testing condition. Specimens tested at the faster loading rate of 250 mm/min are plotted with open markers, while specimens tested at the slower loading rate of 2 mm/min have filled markers. The color red and term ‘dry’ is used to indicate that the specimens were aged at 70 °C in a desiccator, while the blue color and term ‘wet’ indicates aging at 70 °C and 76% relative humidity. The error bars in Figure 5 are the standard deviation of the mean failure load retention divided by the square root of the number of specimens. The failure load decreased after exposure to elevated temperatures, as seen in Figure 5. This is indicative of degradation, although the percent degradation is less than 10% (9.9% for wet fast, 9.0% for wet slow, 8.4% for dry fast, and 5.8% for dry slow), indicating that the material retains most of its failure load after exposure to these aging conditions.

The specimens tested at lower loading rates are much weaker than those tested at faster loading rates. This rate dependence has been seen for other similar composites [5] and is to be expected. Phoenix and Newman developed models for fiber and composite rate dependence and time-dependent properties [36,37]. When the matrix is significantly more complaint than the fibers, as is the case in these UHMMPE laminates, the rate dependence of the composite laminate is predominantly driven by the binder matrix properties compared to the fibers. When the loading rate is sufficiently slow, the matrix has time to creep in shear. Creep in the matrix increases the overloaded length near to a broken fiber(s). Thus, matrix creep properties play a defining role in the composite’s failure load at slow loading rates. In contrast, at fast loading rates, there is no time for the matrix to creep, so only the initial elastic overload length is relevant. This is described, in detail, by Engelbrecht-Wiggans [38,39].

The degradation in failure load, while less than 10%, is statistically significant. Figure 6 shows that with aging, the failure distributions shift to smaller failure loads, indicative of degradation while maintaining the same slope. The constant slope is notable because it implies that the mechanism for failure has not changed.

Figure 6 is plotted as a normal probability plot because the failure distributions for these specimens are well described by a normal distribution, as opposed to a Weibull distribution. Comparable testing performed on PPTA laminates of a similar construction resulted in Weibull-distributed failure load distributions [5]. As the laminate construction is highly similar, it is reasonable to conclude that the different fiber and matrix give rise to the different failure load distributions. UHMMPE fibers are well known to have a low frictional force, and the bond strength between fibers and the matrix is typically much lower than that for PPTA fibers. This gives rise to a much longer load transfer length around a broken fiber in UHMMPE than in PPTA. The Weibull distribution arises from concentrated, local damage, which relies on a shorter load transfer length [38,39,40]. Comparison of the failure videos of the UHMMPE specimens [34] and the PPTA specimens [41] show that the PPTA failures are highly localized, while the UHMMPE ones are not. For the PPTA, the material 5 cm away from the failure site visually appears untested, while for the UHMMPE laminate, there are loose fibers throughout the gauge length, and there are typically several breaks along the length of the specimen. This corroborates that, while the PPTA material failed according to local load sharing, giving rise to a Weibull failure load distribution, the UHMMPE material behaves more according to global load sharing, giving rise to a normal distribution [40]. Hence the failure distributions will be plotted as normal distributions for the purposes of this work.

As mentioned in the Experimental Section, the laminate had visible damage. Figure 7 shows the failure distributions for the unaged and most aged conditions, broken down by the amount of visible damage that was assessed prior to aging. While the specimens were initially placed into five different damage categories, these were grouped together for three different classifications for clarity of display on the figure. In general, for the aging conditions shown in Figure 7 as well as the ones not pictured, there are no significant differences between the failure distributions at the different damage levels. Of all the aging conditions, the dry 335 d specimens show some of the largest differences. However, even after 335 d of aging, the curves overlap, showing that the failure load of the most damaged specimens is greater than that of the only minorly damaged specimens. A lack of correlation between visible damage and failure load is consistent with prior observations [2]. It may be that the only fibers that break due to the visible damage are the weak fibers that would break during loading anyway, thus explaining the invariance of the failure load to damage level.

In addition to the failure load decreasing with aging, the modulus also decreases, as can be seen in Figure 8, where the load vs. strain curves for the unaged material are plotted in black, and the aged curves are plotted in red/pink for the specimens aged in a desiccator and blue for the specimens aged at 76% RH. For the specimens subjected to a slow loading rate, there is a clear separation between aged and unaged curves at higher strains (see Figure 8). This decrease in modulus is less pronounced at the faster loading rate, where there is less time for slip and creep to occur. It is not clear how much of a change in modulus would be seen at ballistic rates.

Standard definitions of modulus require calculating the slope of the stress strain curve in the elastic region. The definition of the elastic region is not immediately clear from these load vs. strain curves as there is no well-defined point where the curve deviates from linearity. To investigate this, cyclic loading tests were performed on unaged and aged material. The results for the unaged material are presented in Figure 9, while Figure 10 is for the aged material. Furthermore, Figure 9 compares the load vs. strain and the load vs. displacement plots for an unaged sample. The video extensometer is limited to capturing 17 frames per second, which adds artefacts to the curve at the locations where the crosshead switches direction. Due to this, Figure 8 and Figure 9 are of load vs. displacement. A load vs. strain version of Figure 9 is presented in the Appendix A.

In Figure 7 and Figure 8, the leftmost column is a plot of the entire test as recorded by the data collection software. The other two columns are of the loading and unloading portions of this curve, with all of the loading portions in the middle column and the unloading in the right column. These loading and unloading curves were shifted to all start at zero displacement to allow for a comparison of how the shape changes as loading progresses. For comparison, Figure 11 plots the final loading and unloading portion of the curves on the same axes for all three materials, again shifted to start at zero displacement.

Elastic deformation is typically defined as deformation that is entirely reversible when the load is removed. The first ten cycles of 50 N to 500 N are completely reversible, almost perfectly superimposing in Figure 9. The first few cycles show some differences, particularly in the load vs displacement plot (as opposed to the strain version) as the specimen tightens on the grips. By the ninth and tenth cycles, however, the overlap in Figure 9 is close to perfect, indicating that the specimen has tightened onto the grips as far as it will for a maximum load of 500 N. Even in this low region (cycling between 50 and 500 N), the material is not perfectly linear, and there is a hysteresis loop. For the next two loadings (with maximum loads of 1.5 and 2.5 kN), the behavior follows the prior loading curves. As the load increases further, however, plastic deformation and viscous behavior start to play a role as each loading curve has a larger gap between it and where the previous maximum load was. Thus, plastic deformation starts occurring at fairly low loads of less than 6.5 kN, even though failure occurs at over 12 kN.

The loading curves mostly superimpose, with better superimposition as the load increases. For each loading curve, the greatest change in the local modulus (focusing on a range of 0.375 kN) increases as the load increases until the maximum load is greater than 6.5 kN. After this load is exceeded, the tangent modulus decreases. In addition, after loading to 6.5 kN, the tangent modulus values are highly similar in each subsequent loading. In contrast, the viscous behavior apparent in the unloading curves changes on each cycle as the unloading lines do not follow the same curve, though there appears to be some sort of lower envelope curve that they converge to as the load drops to zero. The difference in similarity in tangent modulus behavior across multiple loading cycles and a difference in the relaxation time can be seen in all three cyclic figures. The material demonstrates hysteresis, indicative of a viscoelastic or viscoplastic material, and aging results in a more pronounced change to the relaxation and unloading behavior than the loading behavior, as seen in Figure 11, with both aged specimens having a shallower lower unloading curve that steepens and crosses the unaged curve at higher loads. The relevance of the change in relaxation times to ballistic applications (for which this material is intended) is unclear; however, it is clear that the material’s behavior is changing over time in complex ways.

In addition to changes in the failure load and modulus, the failure strain also changes with aging, as can be seen in Figure 12. However, the changes for the failure strain are more complex. The failure strain initially increases with aging at all loading rates and aging conditions, but as aging continued, there became no clear trend for the failure strain. For the fast loading rate, the aged failure strain distribution is sometimes greater than and sometimes less than the unaged distribution, while for the slow loading rate, the strain for the aged material is typically greater than that for the unaged. Some of this variability may be due to measurement error, or it could be that initially the degradation due to aging causes an increased failure strain as only a few fibers are broken; hence, there is simply more slip before failure. At higher levels of aging, there is more damage, so the failure happens sooner, causing either only a modest increase in failure strain or no increase at all.

Throughout this analysis, there have only been minor differences observed in failure load between the specimens aged in the humid and dry environments. This is despite an obvious visual difference in these specimens after aging, with the specimens that were aged in the humid environment turning brown. This implies that the visible signs of aging are occurring in the matrix rather than the fibers themselves. Any differences in the matrix degradation should show up more clearly in the specimens tested at the lower loading rate. Figure 5 shows that the specimens tested at the slower loading rate are seen to have a larger decrease in failure load than the specimens tested at faster loading rates, and the specimens aged at humid conditions for long aging times show a larger decrease in failure load than their dry counterparts, perhaps due to an increased load transfer length in the matrix because of matrix degradation.

At most, the degradation in failure load for any of the specimens is less than 10% of the initial failure load and was observed mostly in the specimens tested at slower loading rates. The prior work [2] shows that the failure load of UHMMPE yarns was more significantly reduced by aging at temperatures similar to those used in this study. At approximately 8000 hr (333 d) of aging at 65 °C, a failure load reduction of 25% was observed, more than twice the amount measured in the current study. This difference could be explained by the fact that the polyethylene in the two studies was from different manufacturers. Thus, the formulations may be slightly different, such as the selection and concentration of antioxidants, which could impact their susceptibility to oxidation. In addition, bare yarns were exposed in the prior study, and in this study, the yarns were encased in a matrix, which may reduce opportunities for oxygen diffusion into the material, thus slowing oxidation. 

Although broadly a decrease in failure load was observed as a function of aging time, there is scatter in the trend, which is expected when sampling from a distribution. Between 50 and 57 specimens were tested in each set to provide relatively high confidence intervals on the mean. While a distribution of failure loads is expected from basic Normal/Weibull principles, there are other considerations that may further complicate comparisons. In particular, the failure load may exhibit spatial variation within the roll. This could be due to damage caused during manufacture or transportation and handling, typically to the edges of the roll or the outermost layers. Certain manufacturing defects also create location-dependent strength if a warp thread is consistently damaged during the lamination process, for instance. The specimens were randomly assigned from the original roll location, controlling for visible damage, so location-dependence in failure load should not be affecting these results. Another consideration is the cutting accuracy with regard to a constant specimen width. Specimens were nominally 70 mm wide. Although variations in width between individual specimens could be as much as 1 mm, this width effect on failure load should be minimal. 

The *V*_50_ ballistic performance measure can be calculated from the tensile properties following either Cunniff [30] or Phoenix and Porwal [31], as described in [32]. The predicted *V*_50_ retention is presented in Figure 13, where mean retention is calculated by scaling the *V*_50_ values at various aging times by the predicted initial *V*_50_. The error bars in Figure 13 are the coefficient of variation (standard deviation divided by the mean), scaled by the mean initial *V*_50_ value. The trends for predicted *V*_50_ do not perfectly follow the failure load degradation (seen in Figure 5) as the *V*_50_ calculation also takes into account changes in the modulus and failure strain. The Phoenix and Porwal method is more sensitive to failure strain, which is why it fluctuates more than the Cunniff method. For all methods, aging conditions, and loading rates, the drop in predicted *V*_50_ is never more than 11%. The largest decrease in predicted *V*_50_ values is seen for the specimens tested at the faster loading rates, with the fast and wet specimens having a minimum *V*_50_ retention of 0.893 and the fast and dry specimens having a minimum *V*_50_ retention of 0.905. In contrast, the specimens tested at a slow loading rate had minimum predicted *V*_50_ retentions of 0.957 and 0.945 for the dry and wet conditions, respectively. There is reason to believe that these predictions may be conservative, based on other research [32], and that the true reduction in *V*_50_ may be even less than that predicted by the Cunniff or the Phoenix–Porwal models.

This section may be divided by subheadings. It should provide a concise and precise description of the experimental results, their interpretation, as well as the experimental conclusions that can be drawn.

## 4. Discussion

The failure load of the flexible UD laminate specimens studied herein degrades with exposure; however, the degradation in failure load is less than 10% after 336 d at 70 °C. This minimal degradation in failure load is encouraging for the field application of this system, which we would expect to be at lower temperatures than that used in this study. This observation is supported by our predictions that the theoretical *V*_50_ of armor made from this material would change by less than 11%. It is important to note that the failure load was not the only parameter affected by aging; the entire stress–strain response changes with aging, especially at slower loading rates. Aging clearly impacts the viscoelastic properties of the UHMMPE laminates, as shown in the cyclic tests, in which they exhibited plastic deformation at low loading, indicating a non-recoverable process. The results also indicated that the addition of humidity to the experimental conditions did not appear to have a major impact on the resulting degradation. While this is not surprising as the material is hydrophobic, it has been postulated that the formation of hydrophilic oxidation products may impact further degradation once they are formed. It is possible that the humid exposure led to slightly more degradation than the dry at the very end of the aging time, but this trend is difficult to distinguish from variation in the measurements with the available data. This result does improve confidence in the prior study on UHMMPE yarns [2], in which only dry conditions were used. It should be highlighted that this study should provide further confidence in the long-term performance of UHMMPE-based armors in the typical use and wear environment for body armor as very little degradation was observed, even at these accelerated conditions of temperature and humidity. One can conclude from the limited degradation observed in these accelerated conditions that the armor is likely to perform as expected during routine service use; however, the testing of used armor with known exposure history would be necessary in order to fully validate the armor performance during use.

In comparing this study to a prior study on PPTA-based flexible UD laminates [5], it is interesting to note that the failure load for the PPTA laminates is Weibull-distributed, while the failure load for the UHMMPE laminates is normally distributed. This is supported by an analysis of the failures and is indicative of a longer load transfer length in the UHMMPE specimens. This also has the consequence that changes in the matrix must be more drastic in the UHMMPE in order to have a measurable effect on the failure load. Of the two materials, after 150 d at 70 °C, the PPTA specimens degraded between 7 and 10%, while the UHMMPE degraded between 5 and 8%. The predicted decrease in *V*_50_ is likewise slightly less for the UHMMPE laminate than for the PPTA laminate.

## 5. Conclusions

The degradation in failure load was less than 10% after 336 d of exposure to the environmental conditions of 70 °C both in a desiccator and at 76 % relative humidity. The expected decrease in ballistic performance, as measured by *V*_50_, is thus less than 11%. This gives confidence in the long term performance of UHMMPE-based armor in typical use and wear environments.

The future extension of this work could pinpoint the causes of the observed degradation in failure load, including examining the chemical composition of the fibers, binder, and interface for changes with aging; determining the effects of additional aging; and establishing whether binder degradation alone leads to reduced *V*_50_ performance with or without delamination of the composite.

## Figures and Tables

**Figure 1 polymers-15-01347-f001:**
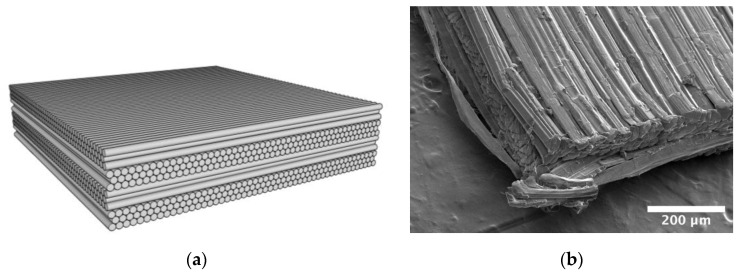
Schematic representation (**a**) of the UD laminate showing the hexagonal packing and the four perpendicular UD layers, each three fibers thick, and a micrograph (**b**) showing the structure of the actual material.

**Figure 2 polymers-15-01347-f002:**
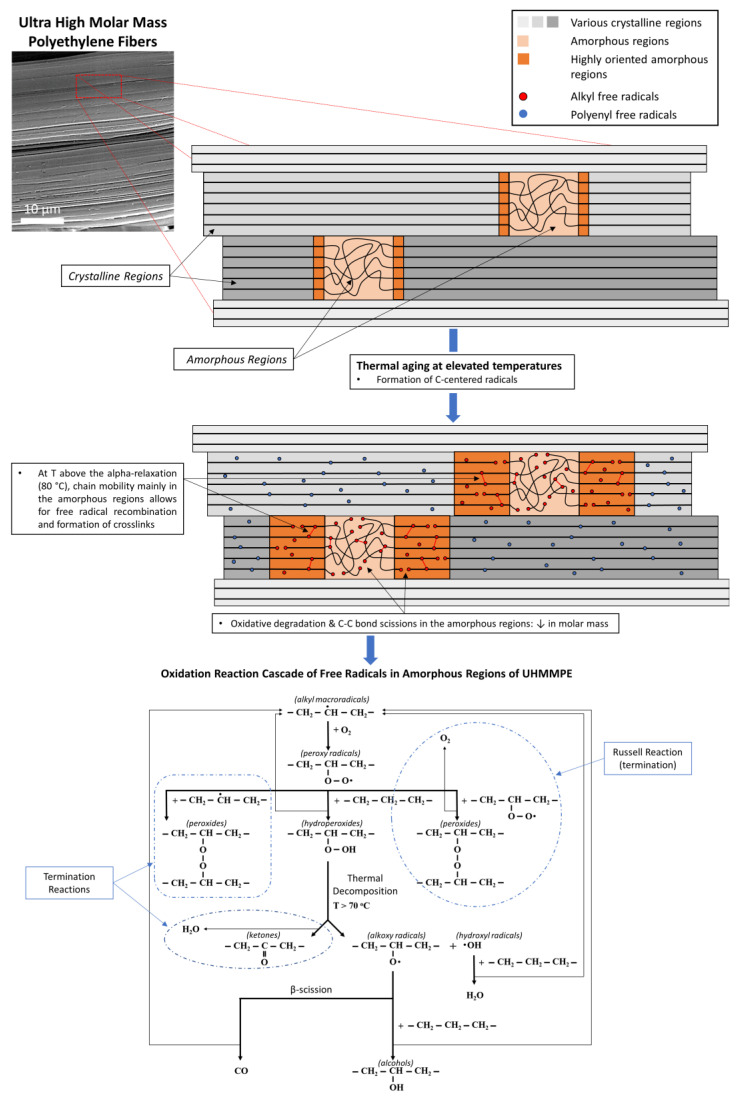
Thermal oxidation of UHMMPE fibers. Schematic of the crystalline and amorphous regions within the polymer fibers; formation of free radicals and the cascade of oxidation reaction in the amorphous regions.

**Figure 3 polymers-15-01347-f003:**
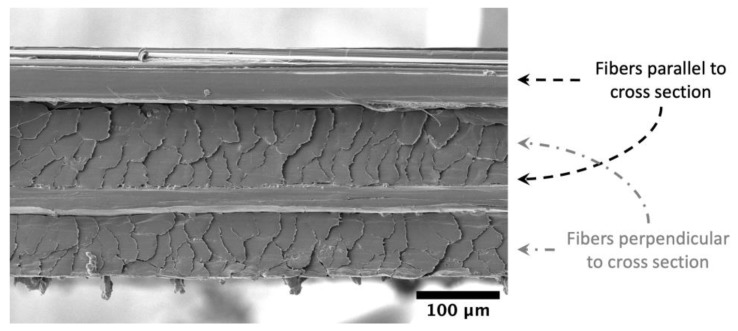
A typical cross-section of the UD laminate using scanning electron microscopy (SEM).

**Figure 4 polymers-15-01347-f004:**
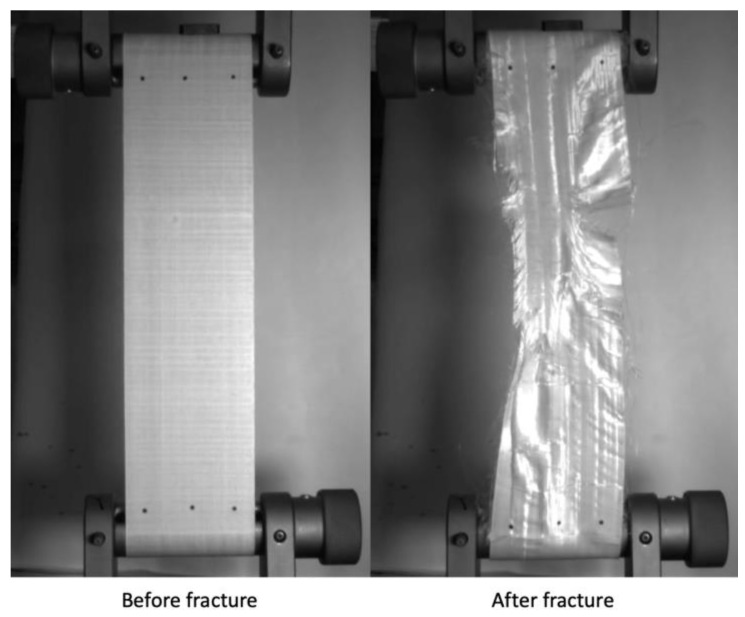
Images of a sample specimen before and after mechanical testing.

**Figure 5 polymers-15-01347-f005:**
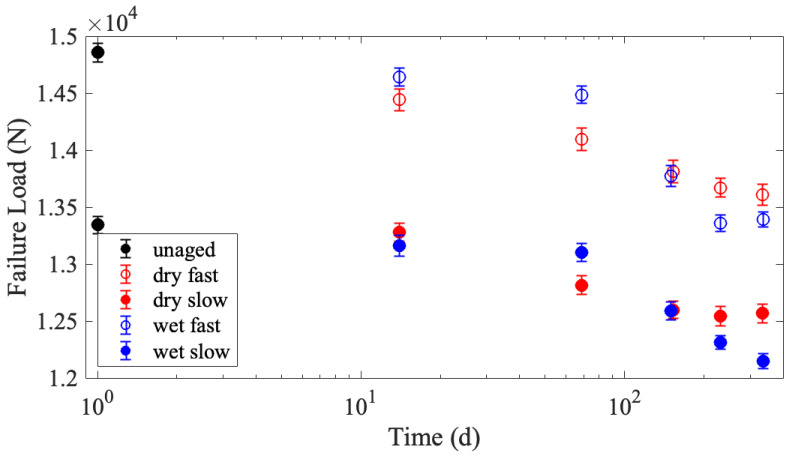
Mean failure load (in Newtons) as a function of days aged, where error bars represent the standard error of the mean (*n* ≥ 50). Specimens are all 70 mm wide with a gauge length of 250 mm, tested at either 250 mm/min (fast–open circles) or 2 mm/min (slow–filled circles). Aging was performed at 70 °C either in a desiccator (dry–red coloring) or at 76% relative humidity (wet–blue coloring).

**Figure 6 polymers-15-01347-f006:**
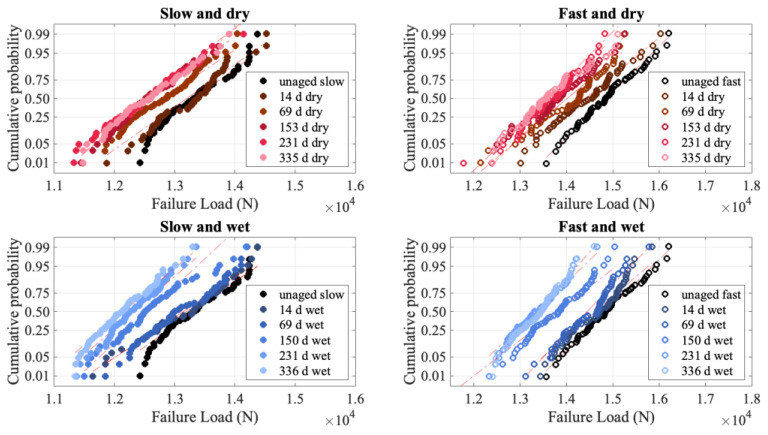
Normal probability plots for failure load.

**Figure 7 polymers-15-01347-f007:**
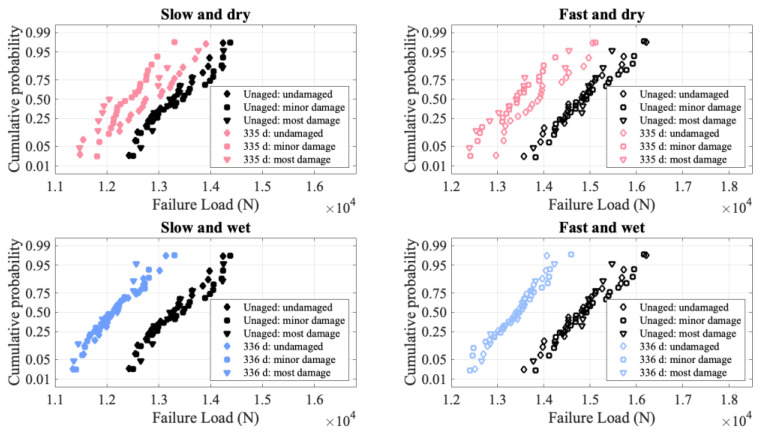
Normal probability plots of failure load (in Newtons) for unaged and most aged material, comparing three categories of damage as visually assessed before aging.

**Figure 8 polymers-15-01347-f008:**
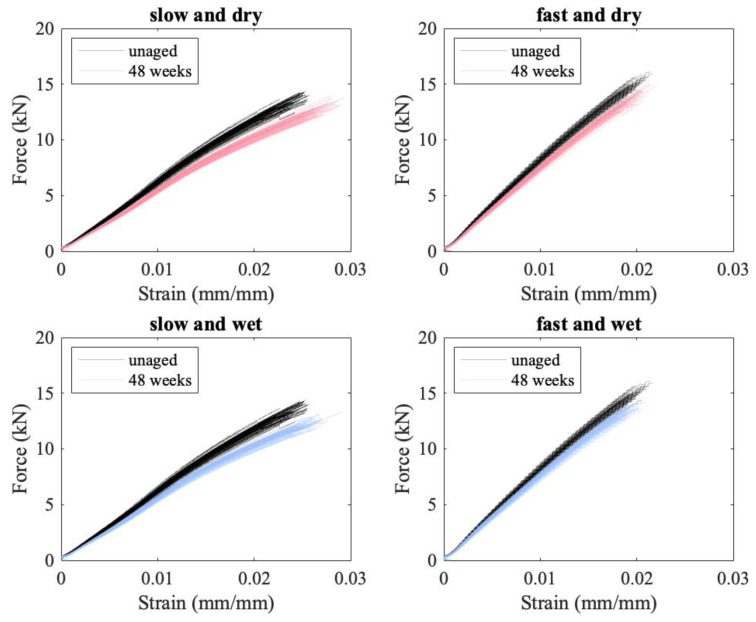
Load vs. strain curves for the unaged and most aged conditions.

**Figure 9 polymers-15-01347-f009:**
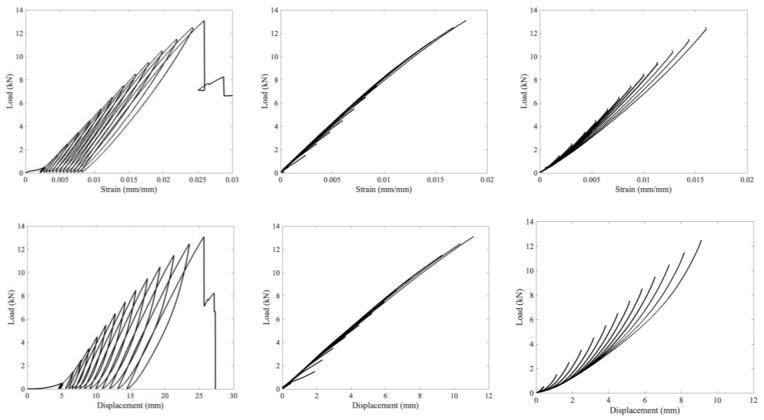
Unaged cyclic-tested specimen with load vs. strain on top and load vs. displacement on bottom. Left plots the whole loading, middle the superimposed loading portions of the curves, and right the superimposed unloading portions of the curves. The curves in the middle and right columns were shifted to start at 0 mm displacement.

**Figure 10 polymers-15-01347-f010:**
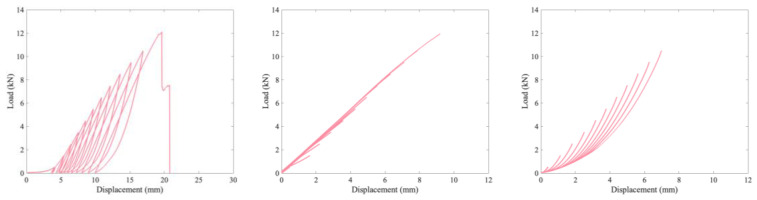
Aged cyclic-tested specimen with 335 d at 70 °C in a desiccator (dry) on top and 336 d at 70 °C and 76% RH on bottom. Left plots the whole loading, middle the superimposed loading portions of the curves, and right the superimposed unloading portions of the curves.

**Figure 11 polymers-15-01347-f011:**
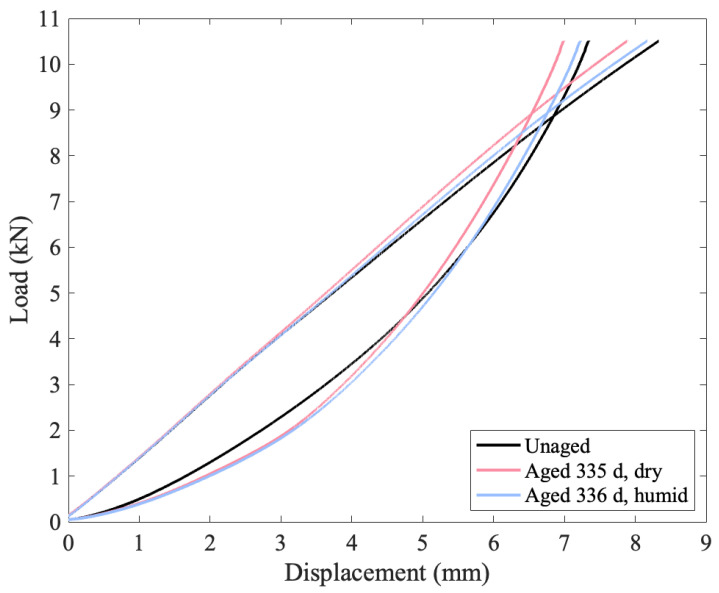
Comparison of the final loading and unloading cycles for all three specimens that underwent the cyclic testing.

**Figure 12 polymers-15-01347-f012:**
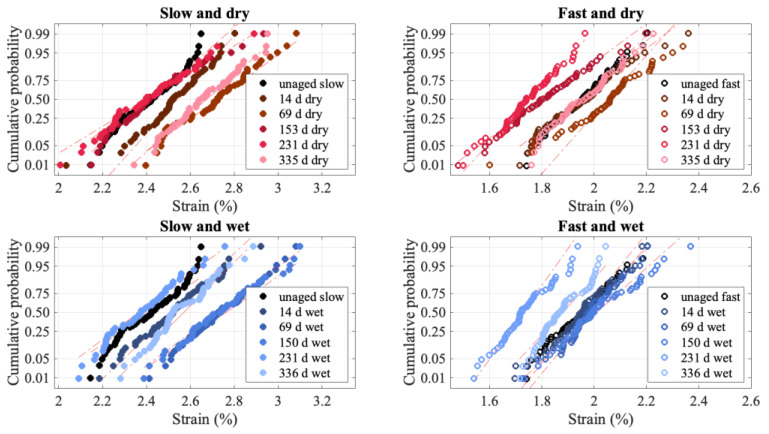
Normal probability plots for failure strain.

**Figure 13 polymers-15-01347-f013:**
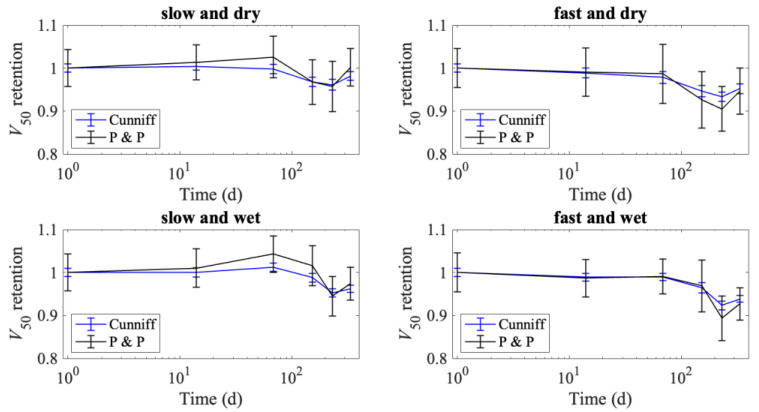
V_50_ retention plots to illustrate the potential impact of aging on the predicted ballistic limit retention for this material.

## Data Availability

The data discussed in this work have been published in [34] and is freely available at https://doi.org/doi:10.18434/mds2-2488.

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
