# Peer review of "Effect of Aging on Unidirectional Composite Laminate Polyethylene for Body Armor"

_polymers, 2023, doi:10.3390/polym15061347_

Round 1

Reviewer 1 Report

The objectives are clear and well defined, in this work was to determine degradation characteristics and behavior ability of composite laminates that comprise multiple layers.

The manuscript is original, and represents a good contribution, addition of knowledge to scientific literature of composite field. Compared with  the unaged control, the treatment with degradation resulted relative modification in mechanical behavior.

The strengths of the method described in the manuscript in the experimental results obtained and correct methodology, but it needs some changes.

The manuscript is well written, but it needs some corrections. It is easy to follow the methodology carried out and the interpretation of the results.

About the change and required information, as follow

-          Tensile tests are used to determine how materials behave under tension load. In a simple tensile test, a sample is typically pulled to its breaking point to determine the ultimate tensile strength of the material. In mechanical tests I prefer to work with normalized values ( Load/Section, in MPa)

-          Photos of fractures

-          Some Incomplete Sentences and Errors:

-the overlap in Figure 8 (left) and Figure 8 (left) is nearly perfect, 291

-as  described in  ¿? 104

-          Lack of information on the characteristics and commercial origin of the materials

-          The article talks about dry and wet. What quantitative difference in absorbed water exists?

Author Response

Thank you for your kind remarks about the manuscript. We have done our best to address the comments and questions that you have raised.

Reviewer 2 Report

The review article titled “Effect of Aging on Unidirectional Composite Laminate Polyethylene for Body Armor” investigated the tensile behavior of an ultra-high molar mass polyethylene (UHMMPE) flexible UD laminate aged for up to 350 d at two accelerated conditions of 70 °C at 76% relative humidity (RH) and of 70 °C in a desiccator.

Here are some points in the current study, that need to address for a better understanding of the research.

1.     Use same referencing style for cross-referencing of Figure 1 and Figure 2 in the literature review.

2.     Please provide the sample development methodology and the matrix type used.

3.     Please provide the labeled image for Figure 3 for more understanding.

4.     Please share that, on what basis the aging conditions were selected? Specifically, for which are of application.

5.     Please provide the images of original specimens before and after testing.

6.     In the caption of Figure 8, “Aged cyclic-tested specimen, with 335 d at 70 ºC in a desiccator (dry) on top and 336 d at 70 ºC and 76 % RH on bottom.” What is the reason of aging 335 and 336 days, the difference is very minor. So, is there any specific reason of this difference?

7.     Provide a concise conclusion at the end of discussion about ultimate aging effect on the final properties of composites.

8.     Literature review is very limited and need to enhance.

9.   There is a lot of work have been done on the performance of reinforced composite. Below are some studies referred to enrich the literature review for this article.

·       Study of dynamic compressive behavior of aramid and ultrahigh molecular weight polyethylene composites using Split Hopkinson Pressure Bar (DOI: 10.1177/0021998316635241)

·       Effect of Different Dielectric and Magnetic Nanoparticles on the Electrical, mechanical, and Thermal Properties of Unidirectional Carbon Fiber-Reinforced Composites (DOI: 10.1155/2022/5952450)

·       Investigation of impact properties of para-aramid composites made with a thermoplastic-thermoset blend (DOI: 10.1177/08927057211021464)

This is an interesting and novel research work, that is very productive for the subject field. Need some changes and addition of further literature review before acceptance.

Author Response

(The authors gave the same response as above.)
